# The History of Pertussis Toxin

**DOI:** 10.3390/toxins13090623

**Published:** 2021-09-05

**Authors:** Camille Locht, Rudy Antoine

**Affiliations:** Univ. Lille, CNRS, Inserm, CHU Lille, Institut Pasteur de Lille, U1019-UMR 8204-CIIL-Center for Infection and Immunity of Lille, F-59000 Lille, France; rudy.antoine@pasteur-lille.fr

**Keywords:** ADP-ribosylation, leukocytosis, histamine sensitization, islet activation, G proteins, pertussis vaccines

## Abstract

Besides the typical whooping cough syndrome, infection with *Bordetella pertussis* or immunization with whole-cell vaccines can result in a wide variety of physiological manifestations, including leukocytosis, hyper-insulinemia, and histamine sensitization, as well as protection against disease. Initially believed to be associated with different molecular entities, decades of research have provided the demonstration that these activities are all due to a single molecule today referred to as pertussis toxin. The three-dimensional structure and molecular mechanisms of pertussis toxin action, as well as its role in protective immunity have been uncovered in the last 50 years. In this article, we review the history of pertussis toxin, including the paradigm shift that occurred in the 1980s which established the pertussis toxin as a single molecule. We describe the role molecular biology played in the understanding of pertussis toxin action, its role as a molecular tool in cell biology and as a protective antigen in acellular pertussis vaccines and possibly new-generation vaccines, as well as potential therapeutical applications.

## 1. Introduction

Pertussis or whooping cough, mainly caused by the Gram-negative organism *Bordetella pertussis*, has long been considered a toxin-mediated disease [1]. In addition to severe cough (translation of the latin word “pertussis”) induced by infection of susceptible individuals with *B. pertussis*, whooping cough patients may also experience pronounced lymphocytosis, as already recognized by Fröhlich in 1897 [2]. Lymphocytosis is associated with severe pertussis disease, mostly seen in young infants. Early reports also described hypoglycemia in children with pertussis [3], and mean plasma-insulin levels were later found to be significantly increased in pertussis patients compared to controls [4]. Furthermore, mice intranasally infected with *B. pertussis* developed a high degree of sensitivity to histamine and died after administration of histamine [5], while non-infected mice are known to be resistant to histamine. These physiological effects of *B. pertussis* infection occur at anatomical sites distant from the respiratory mucosa, the natural niche of *B. pertussis*. With rare exceptions of disseminated infection in highly immune-compromised patients [6,7], *B. pertussis* itself does not disseminate beyond the respiratory tract. Therefore, these remote physiological effects observed in whooping cough patients were concluded to be likely due to soluble factors, termed lymphocytosis-promoting factor (LPF), islet-activating protein (IAP), and histamine-sensitizing factor (HSF), released from the bacterium and reaching anatomical sites other than the respiratory tract.

## 2. From LPF, IAP and HSF to PT

LFP [8], IAP [9], and HSF [10] activities could also be evidenced in mice or rats immunized with whole-cell pertussis vaccines, which accelerated purification and mode of action determination of these factors. After cultivating the Tohama I strain of *B. pertussis*, isolated in the 1950s from a whooping cough patient in Japan, in synthetic liquid medium and harvesting the culture supernatant, Arai and Munoz [11] purified a protein that induced the agglutination of chicken erythrocytes. When left at 2 to 4 °C for 2–3 weeks at a concentration of 800 µg/mL, the protein formed crystals indicating its high purity.

Using the highly purified crystalline protein, Munoz et al. [12] were then able to solve the controversy about the identity of LFP, IAP, and HSF as a single entity. Thus, 0.5 ng/mouse of the crystalline protein sensitized 50% of the animals to histamine, 8–40 ng/mouse induced leukocytosis, and 2 ng/mouse induced increased insulin secretion. In addition, microgram amounts of the material given intraperitoneally caused the death of mice.

Moreover, antibodies against LPF purified by different means, cross-reacted with independently prepared IAP and the crystalline protein. Based on these observations, Munoz et al. [12] concluded that LPF, IAP, and HSF were in fact the same molecule, which they named pertussigen and which is today mostly referred to as pertussis toxin (PT).

## 3. PT as an ADP-Ribosylating Toxin

In mice and rats PT exacerbates adrenergic ß-receptor-mediated insulin secretion and reverses epinephrin-induced hyperglycemia [9]. This could be linked to changes in cAMP levels, as well as calcium movement in pancreatic ß cells, as PT treatment of pancreatic islets augmented cAMP levels and calcium efflux [13]. However, PT alone did not result in elevated cAMP levels, which required the presence of adrenergic ß-receptor agonists together with PT. In fact, Katada and Ui found that membrane-bound adenylate cyclase is inhibited by epinephrine via α-adrenergic receptors in the presence of GTP, and that the inhibition was blocked by PT [14], suggesting that PT modified the mechanism by which the receptor regulates adenylate cyclase activity. Subsequently, they found that a 41-kDa protein of membrane fractions from rat C6 glioma cells was ADP-ribosylated by PT in the presence of NAD and ATP [15], suggesting that PT enhanced the receptor-mediated GTP-induced activation of adenylate cyclase by ADP-ribosylating one of the components of the membrane receptor-adenylate cyclase complex.

This 41-kDa protein turned out to be a guanine nucleotide-binding protein involved in the regulation of membrane-bound adenylate cyclase. It is distinct from another guanine nucleotide-binding regulatory protein, named Ns, a substrate for cholera toxin-mediated ADP-ribosylation, which also alters adenylate cyclase activity [16]. While cholera toxin enhances adenylate cyclase activity via ADP-ribosylation of Ns, the PT substrate protein is involved in the inhibition of the cyclase. The latter was therefore named Ni. These proteins are today mostly referred to as Gs and Gi, respectively. Kurose et al. found that PT-catalyzed ADP-riboslyation interrupts Gi-mediated signal transmission from inhibitory receptors to the catalytic moiety of the adenylate cyclase complex [16].

In fact, the 41-kDa PT substrate was subsequently found to correspond to the alpha subunit of what was initially thought to be dimeric Gi [17], but later shown to be a trimer, composed of the α, β, and γ subunits. ADP-ribosylation of the alpha subunit by PT prevents receptor-triggered GTP binding to Giα and the subsequent dissociation of the Gi subunits, resulting in the loss of hormone-induced inhibition of adenylate cyclase activity.

Subsequently, several additional PT substrates have been identified and shown to be involved in various biological activities. One of them, designated Go, couples fMet-Leu-Phe-binding receptors to phospholipase C, which results in phosphatidylinositol hydrolysis [18], and incubation of human leukemic (HL-60) cells with PT strongly reduced the phospholipase activity. Later, many more G proteins as PT substrates have been identified in various cell types, as well as several subtypes of Gi and Go proteins, which is likely the reason for the diverse biological activities of PT.

## 4. The Structure of PT

In 1978, Kanbayashi et al. [19] described PT as a multimeric protein containing at least three subunits. None of the isolated subunits was biologically active on its own but all subunits had to be combined to potentiate insulin secretory responses. Tamura et al. [20] subsequently refined the subunit structure of PT and identified five subunits. They discovered that heating the toxin with 1% sodium dodecyl sulfate disassembled the five subunits, which could be easily visualized by Coomassie blue staining after polyacrylamide gel electrophoresis. The subunits were named S1 through S5 according to their decreasing apparent molecular weights, ranging from 28 kDa for S1 to 9.3 kDa for S5.

The molecular mass of the assembled holotoxin was estimated by equilibrium ultracentrifugation to be 117 kDa, which was compatible with the 1:1:1:2:1 stoichiometry of the S1:S2:S3:2S4:S5 subunit structure. Incubation of the PT holotoxin with 5 M urea resulted in 4 fractions after carboxymethyl-Sepharose chromatography, corresponding to S5, S1, dimer S3–S4, and dimer S2–S4. The two dimers could be dissociated into their corresponding subunits by incubation in the presence of 8 M urea.

The subunits S2 to S5 assembled as a pentamer, which could bind to haptoglobin-Sepharose, while isolated S1 did not. In contrast, purified S1, but not subunits S2 to S5 expressed ADP-ribosyltransferase activity on Giα. Neither the pentamer nor isolated S1 displayed biological activity, but when they were re-combined, full IAP activity could be recovered in the rat model of insulin response to glucose load. This qualified PT as an oligomeric member of the A-B toxin family, in which the A protomer (in this case the S1 subunit) expresses enzyme activity, and the B oligomer (in this case subunits S2 to S5) is responsible for binding of the toxin to its target-cell receptors. Other well-characterized members of this A-B toxin family include cholera toxin and diphtheria toxin, both also ADP-ribosylating toxins. However, as of today, PT is the most complex bacterial protein toxin known.

The primary structure of the toxin could be inferred from the isolation and sequencing of its structural gene [21,22]. The five subunits were found to be encoded by five cistrons located within a same operon. Each subunit is independently produced as a signal peptide-containing polypeptide, suggesting that they are independently translocated through the inner membrane of *B. pertussis* into the periplasmic space, where assembly into the holotoxin is predicted to occur. All the subunits contain from two to six cysteines involved in intrachain disulfide bonding. The S1 subunit shows significant sequence similarity to the A protomer of cholera toxin, in line with its ADP-ribosylation function. S2 and S3 share approximately 70% sequence similarity, suggesting that their cistrons resulted from gene duplication.

The three-dimensional structure of PT was solved in 1994 at a 2.9 A resolution [23]. The S1 subunit of the toxin has the form of a pyramid resting on a pentameric ring formed by the B-oligomer subunits, from which extrude the N-terminal moieties of the S2 and S3 subunits (Figure 1A). The center of the ring consists of 30 anti-parallel ß-strands, which surround a barrel formed by 5 α-helices. The pore in the center of this barrel is filled by the carboxyl terminus of S1 (Figure 1B). The first 175 residues of the S1 subunit adopt a fold that is common to that of the A subunit of *Escherichia coli* enterotoxin, which is a member of the cholera toxin family. Most secondary structures are conserved between the two toxins. The two cysteines in the S1 subunit are involved in disulfide bonding, as are all the cysteines in the other subunits. Although there is no obvious sequence similarity between S4, S5, and S2 or S3, the two former share similar folds to the carboxy-terminal moieties of the two latter, with two conserved disulfide bonds among all four subunits.

## 5. Receptor-Binding Sites of PT

The binding of the B-oligomer to glycoproteins, such as haptoglobin [20], suggested that the PT receptors might be glycoproteins. This hypothesis was confirmed by the identification of a 165-kDa glycoprotein on the surface of Chinese Hamster Ovary (CHO) cells to which PT-binding via the B-oligomer could readily be observed [24]. Treatment by sialidase abolished PT binding to this protein, and a CHO cell line which lacks the terminal NeuAc->Galß4GlcNAc moiety on glycoproteins was resistant to PT intoxication. These observations indicated that the PT receptors are glycoproteins containing *N*-linked sialo-oligosaccharides, and that the sugar groups are essential for PT binding.

In addition to the 165-kDa glycoprotein on CHO cells, other PT-binding glycoproteins have subsequently been discovered, such as a 115-kDa protein in membrane fractions of erythrocytes [25], a 43-kDa protein in membrane fractions of human T lymphocytes [26] and other sialoglycoproteins on the surface of a variety of cells. In all cases, these proteins could bind to the B-oligomer of PT, and in some cases to the isolated S2–S4 and/or S3–S4 dimers. Interestingly, the two dimers did not show equal binding to the diverse PT receptors. S3–S4, but not S2–S4 bound weakly to the 165-kDa protein of CHO cells, while S2–S4 bound better than S3–S4 to the 115-kDa protein of erythrocyte membranes [25], indicating some level of specificity for each dimer.

Specific modifications in the genes coding for S2 and S3 have identified Asn-105 in S2 and Lys-105 in S3 as critical residues for PT binding to haptoglobin and CHO cells, respectively [27], and the combination of the two mutations resulted in a PT analog that lacked mitogenic activity, which is one of the well-known biological activities of PT. Other alterations in the N-terminal moiety of S2 and S3 also affected receptor binding, thereby identifying this region in the two subunits as the main binding domains. A co-crystal structure of PT complexed with a soluble biantennary undecasaccharide from transferrin containing NeuAcα(2,6)-Gal moieties at each arm showed that the sialogalactose moiety is within hydrogen-bonding distance of Tyr-102, Ser-104, and Arg-125 (Figure 1), common to S2 and S3 [28].

## 6. Mechanism of ADP-Ribosytransferase Activity of PT

PT catalyzes the cleavage of the donor substrate NAD into nicotinamide and ADP-ribose and the transfer of the ADP-ribose moiety to the accepter substrates Gi/oα. Moss et al. [29] found that in the absence of Gi/oα a water molecule could serve as acceptor substrate and NAD-glycohydrolase activity could be readily measured by the release of [*carbonyl*-^14^C]nicotinamide from [*carbonyl*-^14^C]NAD with a *K_m_* for NAD of approximately 20 µM. However, the catalytic rate of G-protein ADP-ribosylation by PT is roughly ten times faster than that of NAD glycohydrolysis in the absence of G proteins [29], suggesting an active role of the acceptor substrate site, identified as the C-proximal cysteine residue located four amino acids from the carboxy-terminus [30], in catalysis.

Both NAD-glycohydrolase and ADP-ribosyltransferase activities require activation of PT by thiol, suggesting that the reduction of at least one disulfide bond is required to either release the S1 subunit from the B-oligomer or to open the active site of the S1 subunit. Later it was found that isolated S1 also requires activation by thiol for the expression of its enzymatic activity [31], indicating that the intrachain disulfide bond has to be cleaved to open the active site of the toxin. In contrast, ATP, another PT-activating molecule, was only required to activate the holotoxin and not the purified S1 subunit [31], suggesting that ATP serves to dissociate S1 from the holotoxin.

The molecular cloning of the PT gene, the determination of its sequence and the production of recombinant S1 in *E. coli* [32] made it possible to use site-directed mutagenesis to identify critical residues involved in the enzymatic activities of S1. Carboxy-terminal deletions of over 50 residues strongly affected ADP-ribosyltransferase activity without modifying NAD-glycohydrolase activity [33], indicating that the NAD-binding site and the catalytic residues are located within the first 180 amino acids of S1 and that the carboxy-terminal region is involved in the recognition of cognate acceptor substrates.

Systematic analyses of the roles of amino acids that are conserved between PT and other ADP-ribosylating toxins, especially cholera toxin and *E. coli* heat-labile toxin [21,22] identified critical residues involved in NAD binding. Modifications of the conserved Arg-9 residue resulted in a dramatic drop in the enzymatic activities of PT, as was found almost simultaneously by three independent groups [34,35,36].

The first catalytic residue Glu-129 of PT was initially identified by photolabeling with [*carbonyl*-^14^C]NAD^+^ again almost simultaneously by three independent groups [37,38,39], followed by site-specific mutagenesis [35,36] and kinetic analyses of the mutant forms [40]. Replacement of Glu-129 by aspartate reduced the enzyme activity by several orders of magnitude but still allowed kinetic characterization of the enzyme activities. While the *K_m_* for NAD was not affected by the amino-acid substitution, its catalytic velocity was reduced approximately 200-fold [40], indicating direct involvement of Glu-129 in catalysis, similar to what had been observed for other ADP-ribosylating toxins. The only additional residue involved in catalysis is His-35, identified by a similar approach [41,42]. This allowed an enzyme mechanism to be proposed [43] in which Glu-129 would retrieve the ribose 2′-OH proton of NAD, thereby destabilizing its nicotinamide-ribosyl bond. His-35 would activate the acceptor cysteine residue of the acceptor substrate G protein to facilitate the nucleophilic attack of the weakened *N*-glycosidic bond, resulting in the release of nicotinamide and ADP-ribosyl transfer onto a water molecule or the cognate Gα protein (for the position of the critical residues, see Figure 1C).

## 7. From PT to PA

Very soon after the identification of the whooping cough agent *B. pertussis* in 1906 [44], efforts were deployed to develop pertussis vaccines. The first vaccines consisted of inactivated whole-cell *B. pertussis* preparations. However, the vaccines were manufactured in many different ways and efficacy varied considerably between vaccines, until standardized whole-cell vaccines and non-clinical potency assays became available in the 1940s [45]. The implementation of these vaccines resulted in an impressive drop in pertussis incidence in countries with high vaccination coverage. However, vaccine-associated adverse events fostered efforts to dissociate the protective antigen(s) (PA) from the toxic elements of the vaccines and to develop more defined, acellular pertussis vaccines (aPV).

Initial attempts to identify PA and to develop aPV date back to the 1950s and made use of the standardized mouse intracerebral *B. pertussis* challenge test to evaluate potency of PA [46]. In this test, potency was found to correlate with protective efficacy in children [47]. Although the exact nature of PA was not identified in these initial studies, there was some evidence that it sensitized mice to histamine to a similar extend as was observed with whole-cell vaccines. That PA was in fact identical to HSF, IAP, LPF and therefore to PT was first proposed by Levine and Pieroni [48] and subsequently experimentally demonstrated by Munoz et al. [12], who showed that as low as 1.7 µg of crystalline PT detoxified by treatment with glutaraldehyde resulted in the survival of 50% of the mice intracranially challenged with *B. pertussis*.

However, controversy persisted. Keogh et al. had shown that *B. pertussis* extracts and culture supernatants can agglutinate erythrocytes from various animal species [49], and that, based on passive transfer experiments, the haemagglutinin (HA) activity was associated with protection in the mouse intracranial challenge model [50]. However, this was not confirmed by others using various HA preparations to actively immunize mice prior to intracranial challenge [51]. Two decades later, Sato et al. found that their purified preparation of PT had HA activity [52] and could in fact identify two molecular entities with HA activity in *B. pertussis*, one corresponding to PT and the other corresponding to a filamentous protein and was therefore called filamentous haemagglutinin (FHA) [53].

Munoz et al. subsequently showed that while µg amounts of glutaraldehyde-treated PT protected mice in the cerebral challenge model, up to 12 µg of FHA failed to protect in this model, unless it was contaminated with trace amounts of PT [54]. Definite proof of PT being the main PA of *B. pertussis* came from studies using highly specific anti-PT monoclonal antibodies. In particular, monoclonal antibodies that neutralized the LPF and IAP activities of PT also protected mice against intracerebral and aerosol challenge with *B. pertussis*, while monoclonal antibodies that did not neutralize these PT activities provided no protection [55]. However, Robinson and Irons showed evidence of a synergistic effect of PT and FHA on protection in the intracranial challenge model [56], and FHA, even in the absence of PT, was also shown to protect mice in a mouse aerosol challenge model [57], suggesting that the combination of both antigens may be required for optimal protection.

## 8. From PT/PA to aPV

The identification of PT as the main PA was the cornerstone in the development of defined aPV. Thanks to the pioneering work by Sato et al., Japan was the first country to implement an aPV composed of formaldehyde-treated PT together with FHA, co-purified from the culture supernatant of the *B. pertussis* Tohama I strain and formulated with aluminium hydroxide [58]. The vaccine had a remarkably improved safety profile over that of the Japanese whole-cell vaccine with comparable efficacy and was implemented in Japan since 1981 for two-year-old children. Following the roll-out of this vaccine, there has been a constant decrease of pertussis incidence in Japan [59], showing high effectiveness of this vaccine to control pertussis.

A first randomized controlled study on aPV was carried out in Sweden, where two Japanese vaccines were investigated and compared to a placebo control [60]. One of the vaccines contained formaldehyde-inactivated PT and FHA, and was thus similar to the vaccine used in Japan, while the other vaccine contained only toxoided PT. The study confirmed the improved safety profile of the aPV over whole-cell vaccines and showed comparable efficacy of about 80% after two doses during a 15-months follow-up against culture-confirmed whooping cough of over 30 days duration. While both vaccines protected equally well against more severe disease, efficacy against milder disease was stronger with the two-component vaccine than with toxoided PT alone. Nevertheless, this study established conclusively that PT was the major PA against pertussis in humans as well.

Most high-income countries have now switched from the whole-cell vaccines to aPV, all of which contain detoxified PT, and most also contain FHA with or without additional antigens, such as pertactin and serotype 2 and 3 fimbriae [61]. In these vaccines, PT is toxoided by several different means, most frequently by formaldehyde and/or glutaraldehyde treatment. A mono-component aPV combined with diphtheria and tetanus toxoids containing hydrogen peroxide-inactivated PT also provided 70–80% efficacy against pertussis with at least 21 days of paroxysmal cough [62] and is currently used in Denmark.

The identification of active-site residues of PT made it possible to engineer genetically detoxified aPV [63]. While chemical inactivation of PT affects the antigenicity of the protein, genetic replacement of Arg-9 by lysine and Glu-129 by glycine had only minimal impact on the recognition by anti-PT antibodies, in particular by PT-neutralizing antibodies [64], and the genetically detoxified PT displays a near identical structure to its wild-type version [65]. Genetically detoxified PT is also more immunogenic in humans [66] and induces anti-PT antibodies of longer duration than chemically detoxified PT [67]. aPV containing genetically inactivated PT have been used in Italy for more than two decades [68] and is currently licensed for immunization of individuals aged 11 years and older in Thailand [67].

Genetic detoxification of PT has also been instrumental for the development of novel, live attenuated nasal pertussis vaccines [69,70]. While aPV have shown their protective efficacy against pertussis disease, several studies have suggested that they do not prevent *B. pertussis* infection and transmission [71,72]. In fact, murine studies suggest that aPV immunization may even prolong nasal carriage by inhibiting the recruitment of IL-17-producing resident memory T cells and neutrophils to the nasal tissue [73]. In contrast, live attenuated pertussis vaccines in which PT has been genetically inactivated have been shown in non-human primates to protect against both pertussis disease and nasal colonization [74,75]. One of these live vaccines is currently in advanced stages of clinical development [76,77] (ClinicalTrials.gov Identifier: NCT03942406). As pertussis is one of the most contagious respiratory diseases [78], it may be important for a pertussis vaccine to provide protection against both pertussis disease and infection in order to ultimately control whooping cough [79]. Mucosal immunization with attenuated pertussis vaccines producing genetically inactivated PT may therefore perhaps be helpful in achieving this goal.

## 9. Conclusions

PT is the most complex bacterial protein toxin identified so far, with respect to both the structure and diversity of biological activities. Before the identification of PT as a single molecular entity, it has been referred to as LPF, IAP, or HSF. In addition, it is the main PA in pertussis vaccines. Some investigators have even suggested that in its toxoided form it is both essential and sufficient for effective vaccination of children and adults [62]. In agreement with this notion, the administration of a single humanized PT-neutralizing monoclonal antibody has been shown to prevent all clinical manifestations of pertussis in neonatal non-human primates [80]. However, it did not prevent colonization of the nasopharynx by *B. pertussis*, in contrast to what was seen after mucosal vaccination with live attenuated vaccines [74,75]. Since *B. pertussis* is a strictly mucosal pathogen, mucosal immunity is likely to be important for the prevention of bacterial colonization [81]. Therefore, in order for a vaccine to effectively protect against both pertussis disease and *B. pertussis* infection/transmission, it should preferably induce potent neutralizing anti-PT serum antibodies, combined with potent local immunity in the upper respiratory tract [82] as the two most crucial immune mechanisms to control whooping cough.

In addition to its important role in vaccines, PT has also been highly instrumental in deciphering a variety of signaling pathways. Compared to other bacterial ADP-ribosylating toxins, for which the receptors and substrates are usually unique, PT binds to a variety of different receptors and ADP-ribosylates many different acceptor substrates of the Gi/o family of signal-transducing proteins. Although it does usually not cause cell death, it can profoundly perturb the physiology of many cell types. For many of them physiological disturbances by PT occur through the interruption of the regulation of cAMP production. However, among others, PT also interferes with potassium or calcium influx, cGMP levels and phototransduction [83]. In vivo, this has been proposed to translate into a wide variety of biological effects, including pulmonary hypertension [84], lung edema [85], brain dysfunction [86], as well as many others, in addition to the leukocytosis, hyperinsulemia, histamine sensitization, and immune cell dysfunction. It may also participate in the cough syndrome, although perhaps indirectly [87]. Since the G protein-coupled receptor family consists of more than 800 members [88], it is likely that in the years to come many additional activities of PT will be uncovered, potentially leading to novel therapeutic applications, such as for the treatment of HER_2_-driven breast cancer for which PT has recently been shown to inhibit breast cancer cell proliferation and migration in vitro, as well as metastasis in vivo in a mouse model [89].

## Figures and Tables

**Figure 1 toxins-13-00623-f001:**
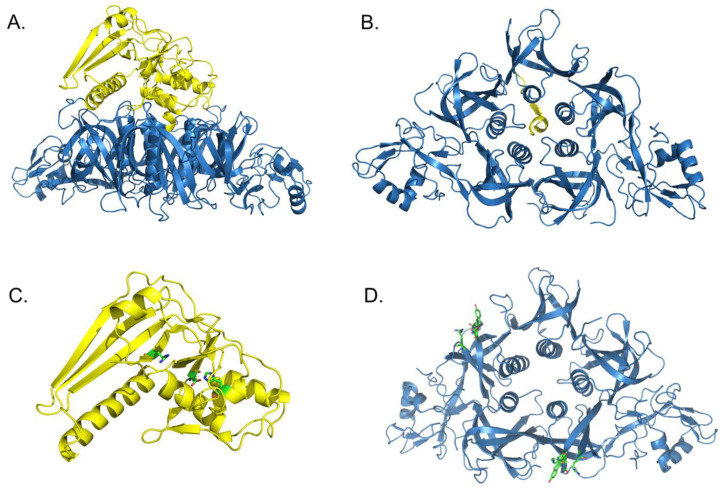
The structure of pertussis toxin. The S1 subunit is depicted in yellow, the B oligomer is in blue. (**A**) Side view showing PT with its S1 subunit on the top of the B oligomer. (**B**) Bottom view showing the B oligomer forming a triangle with 5 α-helices in the middle, surrounding a pore through which the C-terminal end of S1 protrudes. (**C**) Representation of the active side residues Arg-9, Glu-129 and His-35 of S1 in stick mode. (**D**) Representation of the receptor-binding residues Tyr-100, Ser-104, Arg-125 and Asn-105 or Lys-105 (for S2 or S3, respectively) in stick mode. The structures are drawn based on the data from [23].

## Data Availability

No new data were created or analyzed in this study. Data sharing is not applicable to this article.

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
