# Peer review of "The History of Pertussis Toxin"

_toxins, 2021, doi:10.3390/toxins13090623_

Round 1
Reviewer 1 Report
This review presents a concise summary of the discovery of pertussis toxin, its mechanism of action, its functions, and its utility as a vaccine antigen. The review is informative, provides a nice history of research on pertussis toxin, and would be a nice addition to the literature on this important bacterial toxin. My comments are minor in nature.
1, A figure showing the structure of the toxin would be a nice addition and would facilitate visualization of the toxin.
2. lines 286-287: The reference provided demonstrates that the genetically modified PT used in the referenced study induces more persistent antibody levels than the chemically modified PT; however, protection was not measured. Because an antibody threshold for protection that is well accepted has not been established, perhaps "immunity" should be changed to "antibody levels".
3. The manuscript would benefit from additional proofreading. Minor mistakes occur throughout. For example:
line 63, calcium should not be capitalized
line 189, “side” should be “site”
line 213, Ga should be Gα
Author Response
This review presents a concise summary of the discovery of pertussis toxin, its mechanism of action, its functions, and its utility as a vaccine antigen. The review is informative, provides a nice history of research on pertussis toxin, and would be a nice addition to the literature on this important bacterial toxin. My comments are minor in nature.
1, A figure showing the structure of the toxin would be a nice addition and would facilitate visualization of the toxin.
Reply: we have now added a figure on the structure of PT in the revised version.
2. lines 286-287: The reference provided demonstrates that the genetically modified PT used in the referenced study induces more persistent antibody levels than the chemically modified PT; however, protection was not measured. Because an antibody threshold for protection that is well accepted has not been established, perhaps "immunity" should be changed to "antibody levels".
Reply: The reviewer is absolutely right. I have thus changed “immunity” to “anti-PT antibodies”
3. The manuscript would benefit from additional proofreading. Minor mistakes occur throughout. For example:
Reply: Together with the new co-author we have scanned the entire manuscript and corrected the typing errors
line 63, calcium should not be capitalized
Reply: this correction has now been made
line 189, “side” should be “site”
Reply: this correction has now been made
line 213, Ga should be Gα
Reply: this correction has now been made
Reviewer 2 Report
Pertussis toxin has been considered as one important protective antigen and is included in all acellular vaccines in the World. Therefore it is important to introduce its history and biological functions. The design of this manuscript is scientifically proper and logic. The manuscript is well written. I have only a few minor comments. 1. P1, L43, it is better to mention the origin of the Tohama I strain. 2. P1, L45, the Word "affinity" should be taken away. 3. P2, L50-51, it would be more logic to say "0.5 ng/mouse…, 2 ng/mouse… and 8-40 ng/mouse… 4. P2, L63, Calcium efflux should be written as calcium efflux. 5. P7, L30, it should read "in its toxoided form not from".
Author Response
Pertussis toxin has been considered as one important protective antigen and is included in all acellular vaccines in the World. Therefore it is important to introduce its history and biological functions. The design of this manuscript is scientifically proper and logic. The manuscript is well written. I have only a few minor comments. 1. P1, L43, it is better to mention the origin of the Tohama I strain. 2. P1, L45, the Word "affinity" should be taken away. 3. P2, L50-51, it would be more logic to say "0.5 ng/mouse…, 2 ng/mouse… and 8-40 ng/mouse… 4. P2, L63, Calcium efflux should be written as calcium efflux. 5. P7, L30, it should read "in its toxoided form not from".
Reply: 1. I have added that the Tohama I strain came from a Japanese patient with whooping cough. 2. The word “affinity” has been removed. 3. This sentence has been changed. 4. “Calcium efflux” has now been changed to “calcium efflux”. 5. I guess the reviewer meant L309 and requested to change “from” to “form”, which has now been done in the revised version.